# Coagulation Disorders in Sepsis and COVID-19—Two Sides of the Same Coin? A Review of Inflammation–Coagulation Crosstalk in Bacterial Sepsis and COVID-19

**DOI:** 10.3390/jcm12020601

**Published:** 2023-01-11

**Authors:** Georgeana Tuculeanu, Ecaterina Constanta Barbu, Mihai Lazar, Cristina Emilia Chitu-Tisu, Emanuel Moisa, Silvius Ioan Negoita, Daniela Adriana Ion

**Affiliations:** 1Faculty of Medicine, University of Medicine and Pharmacy Carol Davila, No. 37, Dionisie Lupu Street, Sector 2, 020021 Bucharest, Romania; 2National Institute for Infectious Diseases Prof. Dr. Matei Bals, No. 1, Calistrat Grozovici Street, Sector 2, 021105 Bucharest, Romania; 3Department of Anaesthesia and Intensive Care, Elias Emergency University Hospital, No. 17, Marasti Avenue, 011461 Bucharest, Romania

**Keywords:** sepsis, COVID-19, SARS-CoV-2 infection, coagulopathy, immunothrombosis, disseminated intravascular coagulopathy, viscoelastic tests

## Abstract

Sepsis is a major cause of morbidity and mortality worldwide. Sepsis-associated coagulation disorders are involved in the pathogenesis of multiorgan failure and lead to a subsequently worsening prognosis. Alongside the global impact of the COVID-19 pandemic, a great number of research papers have focused on SARS-CoV-2 pathogenesis and treatment. Significant progress has been made in this regard and coagulation disturbances were once again found to underlie some of the most serious adverse outcomes of SARS-CoV-2 infection, such as acute lung injury and multiorgan dysfunction. In the attempt of untangling the mechanisms behind COVID-19-associated coagulopathy (CAC), a series of similarities with sepsis-induced coagulopathy (SIC) became apparent. Whether they are, in fact, the same disease has not been established yet. The clinical picture of CAC shows the unique feature of an initial phase of intravascular coagulation confined to the respiratory system. Only later on, patients can develop a clinically significant form of systemic coagulopathy, possibly with a consumptive pattern, but, unlike SIC, it is not a key feature. Deepening our understanding of CAC pathogenesis has to remain a major goal for the research community, in order to design and validate accurate definitions and classification criteria.

## 1. Introduction

A global estimate on sepsis for the year 2017 found an incidence of 48.9 million cases and nearly 11 million sepsis-related deaths [1]. Along with the epidemiological burden, there are gaps in the early diagnosis strategies, prognostic markers and individualized therapy of sepsis that continue to be problematic [2]. Therefore, sepsis has remained a topic of great interest of both researchers and clinicians for the last decades. Since its emergence in late 2019, the severe acute respiratory syndrome coronavirus 2 (SARS-CoV-2) infection has been diagnosed in over 600 million patients and has caused over 6 million deaths worldwide, according to the World Health Organization [3]. The numerous questions regarding SARS-CoV-2 pathogenesis and therapeutics have been addressed with great participation and responsibility from the scientific community, so that at this point one can attempt to grasp the peculiarities of this systemic viral disease [4]. Dysregulated immune responses, multiorgan failure, vascular involvement and coagulopathy are some of the mutual gross features of coronavirus disease 2019 (COVID-19) and sepsis, but the molecular pathways behind the clinical pictures are not so clearly overlapping [5]. Some authors would rather consider COVID-19 a viral form of sepsis, while others highlight the various distinguishing patterns between the two [5,6]. Since coagulation abnormalities represent an often-fatal complication of both diseases and a number of treatment options are being assessed in this regard, this paper aims to review the current data on the pathogenic sequences responsible for the pathogen invasion (i.e., bacterial pathogen, or SARS-CoV-2) evolving towards an apparent form of coagulopathy. In the first part, the intricate interactions between pathogens, the immune system and hemostasis are outlined. In the last part, these processes are embedded within larger-scale patterns, i.e., the clinically relevant forms of coagulopathy.

## 2. Materials and Methods

Our narrative review was based on the medical literature indexed in PubMed, including in the search process English language articles, from 2000 till 2022. The following keywords were used in the screening process: “sepsis”, “COVID-19”, “SARS-CoV-2”, “coagulation”, “coagulopathy”, “immunothrombosis” and “thromboinflammation”.

After excluding case reports and studies with participants under the age of 18 years old, we selected articles presenting data on coagulation dysregulations in sepsis and/or COVID-19, the pathogenesis of coagulopathy in sepsis and COVID-19, the effector cells of coagulopathy in sepsis and COVID-19, the clinical aspects of coagulopathy in sepsis and COVID-19 and the pathogenesis of disseminated intravascular coagulation (DIC). These aspects were initially searched for in the titles, then in the abstracts, and, when necessary (title and/or abstract were not explicit enough), a full-text evaluation was performed. In cases of further doubts, the supervisor (D.A.I.) was queried.

After the selection strategy mentioned above was performed, thenumber of papers included in our review was 296.

## 3. Molecular Mechanisms of Inflammation–Coagulation Crosstalkin Sepsis and COVID-19

### 3.1. The Triggers—PAMPs/DAMPs/PRRs Interplay

Highly conserved molecular patterns pertaining to pathogens, known as pathogen-associated molecular patterns (PAMPs), and endogenous substances released by cellular injury, called damage-associated molecular patterns (DAMPs), are sought for and recognized by specialized receptors, pattern recognition receptors (PRRs). The interactions between PAMPs, DAMPs and PRRs fire intracellular signaling cascades and/or generate the assembly of inflammasome, which enhances the immune response by producing various cytokines and/or pyroptosis [7,8]. Coagulation abnormalities follow this sequence and contribute to the process called immunothrombosis [9].

#### 3.1.1. Pattern Recognition Receptors

Toll-like receptors (TLRs) and C-type lectin receptors (CLRs) are found on cell membranes and endosomes; conversely, nucleotide-binding oligomerization domain (NOD)-like receptors (NLRs), retinoic acid-inducible gene-I (RIG-I)-like receptors (RLRs) and absent in melanoma-2 (AIM2)-like receptors (ALRs) bind intracellular ligands [7]. In infectious diseases, the common endpoint is the activation of shared signaling pathways: nuclear factor kappa B (NF-κB)signaling, inflammasome signaling, mitogen-activatedprotein kinase (MAPK) signaling and TANK-binding kinase 1—interferon regulatory factor 3 (TBK1–IRF-3)signaling [10,11].

The NF-κB signaling pathway has been commonly associated with the synthesis of various cytokines and chemokines, but, recently, its role in modulating coagulation has become more apparent. The coagulation factor III or tissue factor (TF), factor VIII (FVIII), urokinase-type plasminogen activator (u-PA), plasminogen activator inhibitor-1 (PAI-1), tissue factor pathway inhibitor (TFPI), antithrombin and thrombomodulin have been identified as endpoints in the NF-κB pathway (Figure 1A) [12,13,14,15,16].

Following inflammasome activation, active caspases are generated, promoting the synthesis of proinflammatory cytokines and the cleavage of Gasdermin D (GSDMD), with subsequent pyroptosis (Figure 1A) [8,9]. Caspase 1 and GSDMD-mediated TF activation and release has been reported in murine macrophages following bacterial stimuli and in human monocytic cell lines [17]. Previous studies found that the suppression of inflammasome-mediated pyroptosis in macrophages can diminish TF-induced coagulation disorders [17]. This led to a number of trials investigating the therapeutical value of inflammasome pathway modulators in SARS-CoV-2 infection [18].

#### 3.1.2. Pathogen-Associated Danger Signals

The most-studied PAMPs are constituents of the bacterial wall or membraneand bacterial nucleic acids. Lipopolysaccharides (LPS) are abundant in the outer leaflet of the outer membrane of gram-negative bacteria; their ability to replicate a septic shock pathogenic scenario by binding TLR4 in vitro made them a reliable research tool in the field of sepsis pathophysiology [11]. The SARS-CoV-2 S protein was found to be a TLR4 ligand and NF-κB pathway activator in monocytes and macrophages [19,20,21,22]. SARS-CoV-2 ribonucleic acid (RNA) can activate TLR3 and TLR7 signaling pathways, with interferon α (IFN-α) and IFN-β synthesis in the first 24 h; afterwards, signaling is directed towards proinflammatory cytokine secretion: interleukin 1α (IL-1α), IL-1β, IL-4, IL-6, through the NF-κB pathway (Figure 1B) [23]. Similar to other viral infections, the vigorous cytokine response can often have a systemic impact [24]. The authors suggested that early responses are meant to protect the host, while the subsequent amplification of inflammatory reactions can maladaptively lead to cytokine storm.

#### 3.1.3. Endogenous Danger Signals

The most citedDAMPsare high mobility group box 1 (HMGB1), cell-free desoxyribonucleic acid (cfDNA), histones, heat-shock proteins (Hsp), mitochondria-derived DAMPs and metabolism-associated molecular patterns (MAMPs) [25,26,27].

HMGB1 can bind several PRRs, such as TLR2, TLR4 and TLR9, activating macrophages and endothelial cells (ECs) with the synthesis of proinflammatory cytokines and barrier disruption [28,29,30,31]. HMGB1 modulates fibrinolysis by interacting with plasminogen and tissue-type plasminogen activator (t-PA) and promotes coagulation via TF exposure and anticoagulant protein C pathway inhibition [29,32,33,34]. The role of HMGB1 in sepsis pathogenesis has already been outlined by several studies stating that levels of HMGB1 are elevated in non-survivors, in patients with more severe organ failure and DIC; targeting HMGB1 may improve outcomes in sepsis experimental models [35,36,37,38]. In SARS-CoV-2 infection, HMGB1 is thought to promote acute respiratory distress syndrome (ARDS) development and correlates with disease severity and mortality [39].

Cell-free DNA (CfDNA) activates coagulation via factors XI (FXI) and XII (FXII); accordingly, electron microscopy studies noticed the binding of cfDNA to FXII and high molecular weight kininogen [40]. In this context, the procoagulant activity could be explained by the negative charges and the double-stranded pattern within the DNA structure [41]. The fibrinolytic system is influenced by cfDNA in a concentration-dependent manner: low-physiological levels of cfDNA activate t-PA, enhancing fibrinolysis, while increased cfDNA concentrations determine t-PA inactivation by PAI-1; these processes promote organ damage via microvascular dysfunction in DIC [42,43,44]. Increased PAI-1 levels are thought to be specific of sepsis-associated coagulopathy since they serve as excellent severity markers [45]. Some authors also found increased PAI-1 levels associated withthe cytokine storm of COVID-19, especially IL-6 levels [46]. However, authors comparing the two diseases found significantly higher PAI-1 levels in COVID-19 patients, which could explain the lower D-dimer levels in the COVID-19 group versus the sepsis group in their study [47]. This may limit the plasmin degradation of crosslinked fibrin and the release of D-dimers, explaining the lower levels of D-dimers compared with sepsis found in this study.

Histones exhibit important cytotoxic, proinflammatory and prothrombotic features and have emerged as an important prognostic tool in severe forms of COVID-19 as well as in sepsis; the therapeutic implications for septic patients are currently being investigated [48,49]. Histones promote platelet aggregation and bind prothrombotic molecules such as fibrinogen and adhesion proteins [50,51]. Other interactions concern anticoagulant protein C and tissue factor pathway inhibitor (TFPI) pathways suppression and antithrombin dysfunction through glycocalyx disruption [52,53,54].

#### 3.1.4. How Does SARS-CoV-2 Infect Cells?

SARS-CoV-2 entry in susceptible cells is facilitated by the interaction between spike protein (S) and specific cellular receptors, among which angiotensin-converting enzyme 2 (ACE2) is the most important (Figure 1B). Some other host (co-)receptors, such as CLRs, cluster of differentiation 147 (CD147), phosphatidyl serine (PS) receptors and neuropilin 1 have been cited to contribute to SARS-CoV-2 infection, but none has yet emerged as another essential entry pathway [55,56,57]. However, CD147 was found to initiate proinflammatory signaling via the MAPK pathway, contributing to cytokine storms [58,59]. Host cell proteases—transmembrane protease serine 2 (TMPRSS2) at the cellular surface and cathepsin L within endosomes—are responsible forviral internalization via S protein cleavage [60]. It was shown that activated coagulation FX and thrombin may also cleave S protein and promote viral entry in a potential inflammation–coagulation positive feedback loop (Figure 1B) [61].

Since coagulation disorders have become one of the major systemic effects of SARS-CoV-2 infection, a great amount of research has been focused on viral interactions with the main cellular players of thrombosis—the blood cells. SARS-CoV-2 can infect red blood cells (RBCs) by attaching to Band-3 protein on the erythrocyte membrane; the interaction of S1 spike protein with Band-3 protein leads to alterations in carbon dioxide uptake and oxygen release from hemoglobin, inducing tissue hypoxic changes [62]. Russo et al. found high levels of oxidized glutathione and significant decreases in the enzymes responsible for thereduction inoxidative stress in the RBCs of patients with COVID-19; this led to an increased susceptibility to reactive oxygen species (ROS), resulting in cellular lysis and the inability to carry oxygen [63]. After entering RBCs, SARS-CoV-2 interacts with the 1-betachains of hemoglobin through several viral proteins (ORF1ab, ORF3a, ORF7a, ORF8a and ORF10), causing hemoglobin denaturation and aggravating hypoxia [64]. Hypoxia has been indicated to induce a prothrombotic state and to be associated with venous thromboembolism [65,66]. Shen X.R. et al. reported the presence of viral particles in T lymphocytes and demonstrated that the infection was ACE2/TMPRSS2-independent [67]. Other authors proposed a potential entry pathway of SARS-CoV-2 in T lymphocytes via CD147 [55].

A study by Shen S. et al. revealed a lack of ACE2 expression in both human platelets and megakaryocytes inimmunofluorescence assay (IFA) and Western blot assay testing, suggesting a possible interaction of the viral spike protein with CD147, KREMEN1 and NRP1 [68]. Based on the functions of CD147 in signaling pathways and of NRP1 in cardiovascular, neuronal and immune systems, SARS-CoV-2 interaction with platelets is suspected to regulate the platelet-mediated immune response and promote coagulation dysfunction in COVID-19 [55,56,68,69]. Other authors observed that both ACE2 and TMPRSS2, a serine protease for spike protein priming, are expressed in human platelets, suggesting the possibility that an ACE2-dependent SARS-CoV-2 and platelet interaction is possible, with a consecutive platelet hyperactivation and thrombosis risk in COVID-19 patients [70]. Another study suggests a possible interaction between SARS-CoV-2 viral spike protein and platelet CD42b receptor, thereby promoting interactions with monocytes and cytokine production by monocytes inducing immunothrombosis [71].

Although the interaction mechanisms communicated between platelets and SARS-CoV-2 are not completely understood, may vary among authors and more in-depth studies are necessary, all of the studies performed regarding this point of interest concur to the point that interactions between the virus and platelets are frequent, leading to platelet hyperactivation and thrombosis.

### 3.2. Effector Cells—Monocytes, Neutrophils, Platelets, Endothelial Cells

#### 3.2.1. Monocytes—The Main Source of TF in Inflammation-Induced Coagulopathy

In addition totheir role in conducting responses against foreign pathogens, monocytes promote the activation of coagulation via the extrinsic pathway during inflammatory states; TF is at least partially responsible for this effect [72]. In COVID-19, two functional subsets of monocytes have been identified. Classical, proinflammatory monocytes are numerous in patients with severe disease, while anti-inflammatory monocytes are reduced in these patients and correlate with lung damage markers [73]. This pattern is unlike mononuclear responses in sepsis.

Increasing TF release occurs upon stimulation by LPS, lipoteichoic acid, C-reactive protein or hypoxia (Figure 1A) [74,75,76]. In SARS-CoV-2 infection, platelet–monocyte interactions are a major stimulus for monocyte activation with robust cytokine and chemokine secretion and TF expression [77]. Subsequent to any of these triggers, TF suffers a thiol-disulfide exchange and is turned into the fully procoagulant form by the enzyme called protein disulfide isomerase (PDI) [78]. If fully procoagulant TF is not formed (i.e., the disulfide bond does not form), a stabilized TF/FVII complex, unable to bind FX, interacts with protease-activated receptors (PARs) and it triggers proinflammatory pathways [78]. Therefore, TF release can equally lead to procoagulant and proinflammatory outcomes (Figure 1A).

#### 3.2.2. Activated Neutrophils—At the Intersection of Hemostatic Pathways

Dysregulated neutrophils’ activity in sepsis and septic shock may generate prothrombotic and hyperinflammatory states of varying intensities [79,80]. COVID-19 pathogenesis is increasingly found to rely on a significant contribution from neutrophils. Histological findings were the first aspects pointing in this direction. Autopsies of COVID-19 patients showed important neutrophil infiltrates in pulmonary capillaries [81,82]. In hospitalized patients, bronchoalveolar lavages showed an excessive neutrophil response relative to the other etiologies of pneumonia [83]. On peripheral blood counts, a neutrophil to lymphocyte ratio at admission, as well as its subsequent dynamic changes, have been proposed by several authors to correlate with disease severity, the need for mechanical ventilation and survival [84,85,86,87].

At least three mechanisms could explain neutrophils’ role in coagulation abnormalities: neutrophil extracellular traps (NETs) formation, neutrophil-derived microparticles (MPs) and neutrophil inflammasome activation.

NETs are a key mediator in the maladaptive inflammation–coagulation interplay, associated with sepsis severity, the development of DIC and mortality [88,89,90,91]. The ability of neutrophils to release NETs is one of the most dangerous consequences of neutrophil activation in COVID-19, causing microvascular thrombosis, endothelial dysfunction, tissue damage and eventually organ failure [92,93]. A certain neutrophil population, called low-density granulocytes, is more active in generating NETs and has been identified in greater proportions in COVID-19 patients, enhancing hypercoagulability-mediated organ damage and a poor prognosis [94,95,96].

The stimuli for NETs release are various: PAMPs (i.e., LPS, viral-derived PAMPs), DAMPs, cytokines and chemokines, infection-associated ROS production, activated platelets and ECs (Figure 1C) [97,98,99,100]. SARS-CoV-2-infected neutrophils are able to generate NETs as long as viral replication is underway and with the participation of the ACE2—TMPRSS2 entry pathway or CLRs [101,102,103]. CLRs, together with TLR2, also mediate a specific pathway for immunothrombosis potentiation in COVID-19 consisting of viral-induced platelet activation with vesicle shedding and NETosis [104]. As far as cytokines are concerned, IL-8 promotes NETs release in sepsis as well as in COVID-19, while IL-1β is rather particular for SARS-CoV-2 infection [99,105,106].

The activation of coagulation is mediated through several mechanisms, since NETs not only contain the right triggers but they also create the right conditions for hemostatic processes to take place by engaging platelets, ECs, coagulation factors and inorganic polyphosphate (Figure 1E) [107,108]. The negatively charged web-like surface of NETs (due to histones and nucleic acids) serves as a catalytic medium for the contact pathway activation of coagulation via factor FXII [109,110]. Conversely, interactions between NETs and platelets lead to the activation of the extrinsic pathway; cited mediators are TLR4 signaling, C5a, histones H3 and H4, von Willebrand factor (vWF), fibronectin, P-selectin, HMGBP1 and TF [50,52,108,109,111,112,113,114,115].

Coagulation pathways are further hastened through the inhibition of activated protein C by histones, TFPI degradation by NE and cathepsin G and antithrombin inactivation by NE and cytokine release [52,116,117,118,119,120]. Fibrinolysis pathways also represent a target of NETs components. NE alters plasminogen, mitigating the clot dissolving effects of activated plasmin; moreover, the binding of NE to extracellular DNA protects it from circulating alpha-1-proteinase inhibitor, thus prolonging its procoagulant activity [121,122]. Fibrinolysis resistance can also be explained by the formation of fibrin–DNA tight complexes, which are insensitive to plasmin [114,123,124]. CfDNA activates PAI-1, hindering fibrinolysis via this additional mechanism [42].

Neutrophil activation exposes the endothelium to lytic enzymes—MPO, NE, cathepsin G, ROS and NETs, which can break the functional and anatomical barriers (Figure 1D,F) [125,126]. NETs exert damaging effects on ECs, activating them and promoting more inflammation and thrombosis by TF and adhesion molecules expression, glycocalyx disruption and matrix metalloproteinases activation; accordingly, several study groups found increased levels of syndecan-1 in DIC septic patients [127,128,129,130].

MPsare another mechanism of neutrophil involvement in coagulopathies. The presence of TF in neutrophil-derived MPs has been a subject of controversy [131]. Even though TF in neutrophils could also be acquired from monocytes, several studies managed to induce in vitro TF production in neutrophils with P-selectin or C5a stimulation [132,133]. Currently, it seems that neutrophil-derived MPs can generate thrombin via the intrinsic pathway; interactions between NETs and PS facilitate the assembly of the latter and increase the effectiveness of thrombin generation by MPs [134]. The MPO presence in neutrophil MPs contributes to endothelial barrier disruption and the associated procoagulant phenotype [135].

#### 3.2.3. Platelets’ Function—Much More Than Primary Hemostasis

The last few decades’ research has uncovered the major role of plateletsin coagulation–inflammation crosstalk; unsurprisingly, the amount of evidence of platelets’ involvement in sepsis continues to grow and to get more attractive from a therapeutical perspective. SARS-CoV-2 infection is no exception, especially since micro- and macrothrombotic events are such a frequent finding [136,137,138].

The activation of platelets in infectious circumstances can be caused either by classical endogenous factors or by pathogen-associated exogenous factors—directly or indirectly. Common endogenous platelet activators are adenosine diphosphate (ADP), thromboxane A2 (TxA2), thrombin (acting on PARs), subendothelial collagen and vWF exposure (via glycoprotein VI (GPVI) and glycoprotein Ibα-IX-V (GP-Ibα-IX-V), respectively), and TF and complement component C1q via the C1q receptor (Figure 1D,F) [139,140,141]. Sepsis can associate an acquired type of ADisintegrinandMetalloprotease with ThromboSpondin type 1 motif, member13 (ADAMTS13) deficiency, which generates large circulating vWF multimers, excessively activating platelets and worsening organ dysfunction [142].

Platelets from COVID-19 patients exhibit a particular hyperreactivity to low-dose common agonists (such as collagen, α-thrombin or ADP) compared to healthy controls and other hospitalized patients, including those with ARDS of other etiology, as was found by a number of authors [143,144,145]. The hyperreactivity of thrombocytes was associated withviral RNA entry, increased RNA blood levels and spike protein [70,145,146]. This reduced activation threshold of platelets was proposed by Zaid et al. to explain the hypercoagulable state of COVID-19 patients persisting despite heparin-based anticoagulation regimens [143]. Accordingly, some authors propose additional antithrombotic measures, i.e., antiplatelet agents, following observations of platelet-neutrophil immunothrombi [147]. Moreover, another work of Zaid et al., which found no correlation between D-dimers and platelet number or degranulation markers, suggested that platelet activation may occur independent of coagulation pathways [144].

The exogenous, bacterial-mediated activation of platelets via TLRs has been described [148,149,150,151]. Indirect bacterial–platelet interaction can be facilitated by fibrinogen, vWF, complement and immunoglobulins; IgG and Fcγ receptor IIa may induce platelet activation in staphylococcal and streptococcal infections [152]. In SARS-CoV-2 infection, antibodies are not a key element early in pathogenesis, but considering the widespread propagation of the virus and the high vaccination rates, some authors cite this mechanism to potentially gain more importance in the future [144]. Direct bacterial-mediated platelet activation has been cited in some Staphylococcus species via integrin αIIbβ3 [153,154]. SARS-CoV-2 S protein also contains an integrin binding domain, accounting for another plausible entry pathway [155]. Staphylococcal α-toxin may directly promote thrombosis: it stimulates the activation of platelets via increased calcium influx and factor V (FV) release from α-granules [156].

In SARS-CoV-2 infection, attempts to identify viral RNA in platelets and megakaryocytes haveled to both positive and negative outcomes in different study groups [145,157,158]. The cause of this discrepancy is still unclear; it could be related to different techniques of platelet isolation [159]. Zhang et al. argue that platelets express both ACE2 and TMPRSS2 and that direct viral and protein S binding is responsible forthe subsequent proinflammatory and procoagulant state, mediated through the MAPK pathway (Figure 1F) [142]. However, other studies failed to identify ACE2 and TMPRSS2 mRNA or proteins [144,145]. If and when viral entry is mediated by ACE2 and TMPRSS2, a local disequilibrium of the RAA system is expected, leading to increased TF expression and thrombosis [160,161]. Moreover, Zhang et al. confirmed that SARS-CoV-2 entry promotes ACE2 internalization in platelets, which could eventually explain the conflicting results of other authors regarding ACE2 expression [70]. Alternatively, viral entry could also be mediated by CD147, KREMEN1, neuropilin 1, heparan sulfate, sialic acid or MPs, the latter of which were found to trigger vigorous prothrombotic granules’ release, possibly followed by cell death [158,162]. Since a definitive answer is still lacking in this regard, the main culprit of platelet hyperreactivity—either viral genome or external factors—cannot be clearly identified.

Upon activation by various ligands in sepsis, platelets not only enhance their prothrombotic functions but also gain a particularly proinflammatory phenotype. Despite being anucleated, platelets can assemble NLR family pyrin domain containing 3 (NLRP3) inflammasome and produce IL-1β through IL-1β pre-mRNA processing following LPS-TLR4 stimulation [163,164,165,166,167]. IL-1β is unique, as it generates an autocrine activation loop, with further platelet activation and subsequent IL-1β synthesis; it is packed in platelet-derived MPs and it promotes megakaryocyte maturation [164,165,168,169]. In a comparable way to IL-1β, TF pre-mRNA is spliced to TF mRNA upon activation and shed as active TF [170,171]. The immune response to SARS-CoV-2 infection is characterized by high innate immunity reactivity with excessive and various cytokine release—IL-6, IL-2, IL-4, IL-13, IL-12, tumor necrosis factor *α*(TNF-α)and IL-7 [172]. Among them, IL-6 and TNF-α can directly activate platelets; both IL-6 and IL-1β can prime platelets before stimulation by classical agonists (Figure 1F) [160]. Following activation and/or priming through the aforementioned modalities, platelets release their α and dense granules and a variety of stored cytokines as shown by the experimental detection of increased FV and FXIII and platelet factor 4 release, CD40L, P-selectin exposure and IL-1 de novo synthesis (which partially mirrors the positive feedback mechanism in sepsis) [70,144,145].

Platelet–leukocyte interactions are initiated by the corresponding receptors’ pairing following activation: P-selectin and PSGL1 on neutrophils and monocytes, integrin αIIbβ3 or GPIb, through fibrinogen and macrophage 1 antigen (Mac-1) fibrinogen receptor on neutrophils (Figure 1C) [112,173,174,175,176]. Several studies found increased platelet–monocyte, platelet–neutrophil and platelet–T-cell interactions in COVID-19 patients [145,177]. These interactions lead to MPs release, TF exposure and eventually fibrin formation [145,178]. Significant correlations with inflammatory markers and disease severity were found by several authors [179,180].

Microparticle sheddingis another consequence of thrombocyte activation during sepsis. TLR4 signaling and GPIV ligands are some of the main triggers [165,181]. MPs contain IL-1β and can expose TF and binding sites for coagulation factors (particularly vitamin K-dependent coagulation factors) (Figure 1D,F) [165,182,183]. Their role in COVID-19 pathogenesis is becoming more and more apparent as well. Substantiating the prothrombotic potential of MPs in COVID-19, PS and TF have been identified in their composition; a connection with systemic thrombin formation and pulmonary disease severity was subsequently inferred [144,184,185,186]. Accordingly, markers of platelet activation were found in tracheal aspirates from patients with severe COVID-19 [180]. Other effects of increased MPs release are enhanced bacterial clearance andthe carrying of DAMPs, lipid mediators and platelet-associated receptors (i.e., P-selectin and integrin αIIbβ3) to other membranes [166,187,188,189]. Platelet-derived MPs have been associated with altered prognosis in sepsis due to myocardial dysfunction and the apoptosis of vascular smooth muscle cells and endothelium [190,191,192]. Apoptosis and endothelial cell dysfunction are frequent findings in various scenarios of increased blood levels of inflammatory cytokines—TNFα, IL-6 and CRP—due to persistent immune activation [193,194]. In SARS-CoV-2 infection, severe forms were associated with a lower number of active, PS-containing MPs (versus non-severe forms), possibly due to consumption or sequestration in certain organs [144]. Authors suggested that, since inflammation markers correlated better with MPs concentrations than D-dimers, inflammatory processes and not thrombosis arewhat primarily impact MPs dynamics [144].

In the septic proinflammatory environment, a new generation of thrombopoietin-independent thrombocytes is formed within the bone marrow [72,195]. This inflammation-induced megakaryopoiesis, stimulated by IL-6 and IL-3, leads to a different lineage of megakaryocytes and thrombocytes, which contains more TLRs and IL receptors, produces more IL-6 and TNF-α and exposes the major histocompatibility complex class II (MHC class II), facilitating antigen presentation [196,197,198,199,200,201]. Megakaryocytes are increasingly cited contributors to SARS-CoV-2 infection pathogenesis as well. Unlike platelets, ACE2 and TMPRSS2 protein expression in megakaryocytes is stated with more certainty [70]. As a consequence, pulmonary residing megakaryocytes are theoretically susceptible to SARS-CoV-2 infection and may transfer viral particles when circulating platelets are generated [144]. The diverse repertoire of cytokines and chemokines released by platelets upon activation can also be transferred during megakaryopoiesis. Furthermore, SARS-CoV-2 can locally enhance the production of this new generation of thrombocytes; in line with this theory, increased thrombopoietin levels were found in COVID-19 patients [160,202].

#### 3.2.4. Endothelium Coordinates Both Inflammatory Responses and Coagulation

The endothelium represents an autonomous “organ”, whose integrity helps preserve a specific homeostatic state regarding hemostasis and inflammation.The activation of ECs is a proven key element of sepsis pathogenesis. Accordingly, several endothelium-derived markers serve as diagnostic tools or therapeutic targets [203,204,205,206]. In COVID-19, the first studies pointing to the major role of endothelial dysfunction focused on the histological findings of endotheliitis, a disrupted endothelial barrier, thrombosis and the microangiopathy of alveolar capillaries [207].

The activation of ECs in sepsis is mainly due to the binding of TNF-α and thrombin to TNF receptors and PAR-1, respectively; bacterial LPS and HMGB1 can also play a secondary role (Figure 1D) [72,208]. However, the primary insult in COVID-19-associated endotheliopathy is represented by the direct SARS-CoV-2 entry via ACE2 and TMPRSS2 (Figure 1F) [209]. Both viral particles per se (i.e., S protein) and the associated reduction in ACE2 expression contribute to the activation of ECs and to the subsequent shift towards a proinflammatory and prothrombotic phenotype (Figure 1F). Currently under researchers’ scrutiny, the pathway of angiopoietin 1 (Angpt1)—tyrosine kinase with immunoglobulin-like loops and epidermal growth factor homology domains-2 (Tie2) represents a major factor in ECs homeostasis, particularly when facing inflammation-derived prothrombotic tendencies [206]. Tie2 signaling alterations with excessive angiopoietin-2 release could lead to a phenotype switch in both COVID-19 and the early phases of sepsis [206,210].

A unique feature of sepsis endotheliopathy is the impact of mechanical insults. Shear stress has been indicated by some studies to maintain an anti-inflammatory and anticoagulant state via the NF-κB pathway [211,212]. Therefore, authors have hypothesized that, if adequate shear stress and ECs function were to be preserved, sepsis outcomes could benefit from this anticoagulant and anti-inflammatory phenotype and from the subsequent attraction of endothelial cells progenitors [213].

Regardless of the initial trigger, ECs acquire a proinflammatory phenotype mediated through classical signaling pathways—NF-κB and MAPK [208]. The transcription of adhesion molecule genes (such as intercellular adhesion molecule 1 (ICAM-1), vascular cell adhesion molecule 1 (VCAM-1) and E-selectin), ROS synthesis and matrix metalloproteinases release may result from direct cytokine activity, ACE2 down-regulation or S-protein-related damage (Figure 1F) [178,208,214,215,216,217]. The subsequent increased cytokine production (TNF-a and IL-6 in particular) further promotes endothelial dysfunction: IL-6 contributes to vascular permeability and TNF-a worsens glycocalyx disruption in both diseases [218]. In COVID-19, excessive complement activation via the lectin pathway and a particular hyperactive state of the KKS pathway stand out, increasing vascular permeability [216,219,220].

Following this proinflammatory response, procoagulant mechanisms are subsequently engaged. NF-κB activation determines Weibel–Palade bodies’ exocytosis, with the release of vWF multimers, FVIII and P-selectin [178,217,221,222,223]. The release of vWF multimers in the context of sepsis-associated ADAMTS13 deficiency determines microvascular thrombosis with an impact on organ dysfunction severity (Figure 1D) [142,224]. The protein C anticoagulant system becomes dysfunctional in sepsis due to impaired protein C synthesis and activation, via thrombomodulin and endothelial protein C receptor deficiency [222]. Conversely, activated protein C has been found to mediate inflammation–coagulation crosstalk, similarly to the thrombin-related activation of PARs [225,226]. Other procoagulant modifications found in sepsis, as well as in COVID-19, are reduced TFPI and t-PA synthesis and increased PAI-1 expression [178,208,218,223,227].

Endothelial glycocalyx has become a major link in endothelial dysfunction pathogenesis in multiple diseases. Sepsis-associated glycocalyx injury is initiated through the lytic activity of specific enzymes: glucuronidases (i.e., heparanase), hyaluronidase, plasmin, cathepsin B, proteases and by reactive oxygen species (ROS) (Figure 1D) [226,228]. Hypervolemia (mostly determined by aggressive fluid resuscitation within sepsis management) has been cited to be a glycocalyx-disrupting factor; atrial natriuretic peptide may be incriminated in pathogenesis [229]. Following glycocalyx disruption, various adhesion molecules are expressed, leukocyte transmigration is facilitated and platelets are attracted to the injury site, initiating microthrombus formation [230]. The anticoagulant roles of glycosaminoglycans are therefore compromitted with resulting TFPI and antithrombin dysfunction [231]. In COVID-19, increased syndecan-1, heparan sulfate and hyaluronic acid levels, which correlated with disease severity, point towards significant glycocalyx disruption, but its role in SARS-CoV-2 coagulopathy remains a topic of on-going research [232,233].

A comparative summary of the major commonalities between sepsis and COVID-19 regarding all molecular and cellular mechanisms is found in Table 1.

## 4. From Theory to Practice—Clinical Aspects of Coagulopathy in Sepsis and COVID-19

### 4.1. Progression of Coagulation Disorders—The Pathways towards DIC

As was highlighted in the paragraphs above, septic patients have numerous theoretical reasons to develop nearly any type of coagulopathy. Accordingly, some authors state that practically all patients fulfilling sepsis criteria suffer from a certain type and severity of coagulopathy [238]. In COVID-19, describing and defining the associated coagulopathy (CAC) has been a matter of intense research ever since the global spread and the severe course of SARS-CoV-2 infection became apparent.

#### 4.1.1. What Is DIC?

DIC is defined as “an acquired syndrome characterized by intravascular activation of coagulation with loss of localization arising from different causes; it can originate from and cause damage to the microvasculature, which if sufficiently severe, can produce organ dysfunction” [239]. From a pathogenic standpoint, sepsis-associated DIC is a form of suppressed fibrinolysis-type DIC: the very subtle rise in fibrinolytic activityis highly disproportionate compared to the uncontrolled activation of coagulation (due to abnormally high levels of PAI-1, a reduction in t-PA, TFPI and protein C system activity), a process called “fibrinolysis shutdown” [44,240]. A vicious circle, with continuously worsening hemostasis dysfunction and consequent inflammatory maladaptive responses, leadsto the final stages of DIC, substantially worsening the prognosis for septic patients [241,242].

Within the clinical setting, DIC can take the form of a wide spectrum of manifestations, both manifest and subclinical, sometimes progressing towards a common classical form of consumptive coagulopathy. Sensible markers of coagulation activation and fibrinolysis such as thrombin-antithrombin complex (TAT) and plasmin-α_2_-plasmin inhibitor complex (PIC) can identify the clinically silent forms of coagulopathy; standard coagulation tests showing slight modifications are characteristic of early forms of septic coagulopathy; eventually, completely impaired coagulation panels are found in what is defined as overt DIC [239,243,244]. Overt DIC criteria, despite having high sensitivity and specificity, diagnose advanced stages of coagulopathy, when therapeutic interventions are no longer beneficial. Updated criteria of the International Society on Thrombosis and Haemostasis (ISTH)on DIC diagnosis in sepsis introduced the concept of early-stage DIC in septic patients, namely sepsis-induced coagulopathy (SIC) [245]. The SIC score is only composed of platelet count, prothrombin time (PT) and sequential organ failure assessment (SOFA) score; it remains the most widely employed tool for sepsis coagulopathy assessment [245].

#### 4.1.2. Is Coagulopathy of COVID-19 a Form of DIC?

One of the first important steps towards understanding CAC pathogenesis was launching the hypothesis that coagulopathy in COVID-19 could mirror the natural evolution of the infection, thus displaying an initial respiratory limited phase, possibly followed by a disseminated, systemic phase [5]. This two-phase progression of the disease differs substantially from that of bacterial sepsis. In fact, it appears that severity scores for bacterial sepsis (i.e., SOFA score) do not perform as well in predicting COVID-19 outcomes [246,247,248].

Some authors advanced the term “Pulmonary Intravascular Coagulation” to depict the phenomena behind COVID-19 acute lung injury [249,250,251]. Alveolar fibrin deposits represent one of the main histological markers of COVID-19; therefore, the disruption of the alveolar–endothelial barrier and microvascular thrombosis due to unbalanced activity of coagulation and fibrinolytic systems are factors thought to underly the pulmonary histological features of SARS-CoV-2 infection [252,253,254,255]. An accelerated fibrin turn-over is characteristic of the pulmonary phase of CAC with both endothelial PAI-1 and alveolar (u-PA) excessive release, followed by intense D-dimers formation [256,257]. By contrast, this phase associates a systemic activation of the coagulation cascadewith “fibrinolysis shutdown”—similarly to SIC—as found by viscoelastic testing, despite the overwhelming increase in serum D-dimers [234,251,255,256,258]. In fact, several authors have recently pointed out that D-dimer levels are only associated with fibrin formation dynamics, and high D-dimers do not necessarily imply systemic hyperfibrinolysis; therefore, in COVID-19, evidence points towards an extravascular—i.e., pulmonary—source of D-dimers [259].

A clinically relevant systemic form of coagulopathy can eventually develop in SARS-CoV-2 infection. In advanced stages, it can take the highly fatal form of overtDIC—higher DIC scores (in accordance with ISTH) were found with a high prevalence of up to 70%, in COVID-19 non-survivors [260]. However, the systemic phase of CAC can take different intermediary forms before progressing towards overtDIC. In the earliest stages, probably overlapping the pulmonary phase, a trend towards platelet- and fibrinogen-mediated hypercoagulability is shown by viscoelastic assays [255]. A slight prolongation of PT and activated partial thromboplastin time (aPTT) can occur and this anomaly remains subtle for a longer period of the disease course, possibly due to FVIII and fibrinogen synthesis within the acute phase response [257]. An increased thrombin generation potential can be identified in both sepsis and SARS-CoV-2 infection; this enhanced activation of coagulation is maintained in patients with COVID-19 despite prophylactic anticoagulation, unlike septic patients [147]. Moreover, the pathologic changes in plasma thrombin and plasmin generation found in this study permitted enhanced fibrin formation in both COVID-19 and sepsis, with unexpectedlydelayed times to thrombin, plasmin and fibrin formation in COVID-19, suggesting a loss of coagulation-initiating mechanisms in severe COVID-19 [147]. The platelet count remains only slightly decreased until an overt DIC stage is reached and, unlike SIC, it is not so clearly associated with mortality [257,261].

Other cited forms of coagulopathy that have been described in association with SARS-CoV-2 infection are a form of hemolytic microangiopathy due to ADAMTS13/vWF disequilibrium, vasculitis linked to antiphospholipid and anti-β_2_-glycoprotein antibodies, and possibly hemophagocytic lymphohistiocytosis [235,236,257,262].

Considering the aspects reviewed in the paragraphs above on SIC and CAC pathogenesis, we depicted numerous shared mechanisms when it comes to coagulation–inflammation interplay. Nevertheless, in a more in-depth assessment, subtle differences become apparent and the resulting clinical picture does not display as many similarities between the two diseases.

Overall, CAC is a very complex form of coagulopathy, displaying two anatomically and pathogenically different phases; even though it could at times fulfill DIC definition and criteria, other factors should be considered before setting the equal sign between the two. Consequently, a collective effort is being made to accurately define and classify CAC; a systematic approach is provided by Ibaet al. in one of their latest reviews [5].

### 4.2. Searching for the Proof—Coagulation Studies

#### 4.2.1. Standard Coagulation Tests

***Typical laboratory findings in SIC and sepsis-associated DIC.*** ISTH diagnostic criteria for SIC and sepsis-associated overtDIC employ the most important markers in sepsis coagulopathy, with high availability in clinical circumstances: platelet count, D-dimers or fibrinogen/fibrin degradation products (FDP), PT, serum fibrinogen levels and SOFA score [245]. Thrombocytopenia is very frequent in sepsis and serves as a marker of poor prognosis [237]. A very low or rapidly falling platelet count can be a typical sign of DIC and it predicts the length of intensive care unit (ICU) stay, mortality and increased risk of bleeding [263]. Increased D-dimer levels are a frequent finding in sepsis-associated DIC [264]. High levels of D-dimers are associated with 28-day mortality in patients with infection or sepsis [265], a level of 4μg/mL or more inducing a 12.6-fold increase in mortality in patients with severe sepsis [266].However, several authors [264,267,268] have implied that D-dimers do not accurately correlate with mortality. For instance, a sepsis subgroup, with advanced illness and overtDIC criteria but particularly low D-dimer levels showed very high mortality rates [268]. There are also authors that support both situations, demonstrating that both low and extremely elevated D-dimer values are associated with a higher risk of death in patients with sepsis [269]. Both COVID-19 and sepsis are associated with the endogenous activation of coagulation and fibrinolysis, presenting elevated fibrinogen, D-dimers, solublethrombomodulin and plasmin–antiplasmin complexes. [147]. Although COVID-19 and sepsis patients presented significantly high levels of D-dimers and TF-containing MPs’ activity compared to healthy controls, the serum levels of D-dimers were significantly lower in patients with COVID-19 compared to patients with sepsis [47]. Patients with COVID-19 also presented high levels of active PAI-1 (whereas patients with sepsis did not), which may limit the plasmin degradation of crosslinked fibrin and the release of D-dimers, explaining the lower levels of D-dimers compared with sepsis [47].

The “fibrinolysis shutdown” that occurs in sepsis due to increased PAI-1 synthesis has been incriminated in this scenario [267]. This hypothesis is supported by corresponding lower PIC levels in these patients [268]. Standard coagulation tests PT and aPTT cannot accurately measure in vivo hemostatic processes; however, when prolonged due to excessive consumption, they correlate with DIC severity and bleeding risk [263]. Among clotting factors, FVIII and fibrinogen are considered less valuable in assessing DIC: FVIII is frequently increased due to excessive release together with vWF and fibrinogen, being an acute phase protein that maintains a close-to-normal serum level until the very advanced phases of DIC [270,271].

***What stands out in CAC?*** There are at least two aspects that distinguish SIC and CAC clinical pictures: D-dimers and platelet count. As was highlighted before, in COVID-19, D-dimers are thought to originate from a process of intense fibrin formation and subsequent degradation within alveoli, in the context of pulmonary intravascular coagulation [249,272]. D-dimers in COVID-19 correlate with poor prognosis [258,273]. They have particularly high levels even in early phases, much higher than in classical DIC settings, and are not initially associated with consumptive coagulopathy markers such as prolonged standard coagulation tests or hypofibrinogenemia [274,275,276,277]. Regarding platelet count, a definitive conclusion is yet to be proposed. Some authors found worsening thrombocytopenia with increased COVID-19 severity, but plenty of studies showed that, even though mild thrombocytopenia (100–150 × 10^9^/L) is frequent, platelet counts lower than that are identified in a very small group of patients [261,278,279]. This could be explained by a compensatory response of platelet production [280]. Overall, further investigations are needed to determine whether low platelet count is a prognostic marker in COVID-19 and whether it arises mainly due to the viral infection itself or to other factors as well (antiphospholipid syndrome, immune thrombocytopenic purpura, hemophagocytic syndrome, heparin-induced thrombocytopenia, drug-induced myelosuppression, etc.) [251].

Another distinctive feature has been outlined by the major work of Tanget al.,who followed the dynamics of coagulation markers in COVID-19 patients [276]. It was shown that COVID-19 coagulopathy, as it can be described by standard coagulation tests, is a continuously evolving process. Specifically, after approximately 7–10 days from admission, non-survivors underwent a massive shift in their biological picture—from slightly increased PT and aPTT, high D-dimers and near-normal fibrinogen (the characteristics of a suppressed-fibrinolytic-type DIC) towards extremely prolonged PT and aPTT, extremely high D-dimer levels and very low fhbrinogen levels (the characteristics of an enhanced-fibrinolytic-type DIC) [276,281]. The Histological findings of alveolar hemorrhage along thrombosis in the autopsies of COVID-19 patients could be a manifestation of this profile switch of CAC, as was suggested by Asakuraet al. [281,282].

#### 4.2.2. Advanced Coagulation Studies—Viscoelastic Tests in SIC and CAC

Viscoelastic tests (VETs), i.e., thrombelastography and thromboelastometry, are advanced techniques, using whole blood to evaluate in real time the dynamics of clot formation and the distinct input of platelets, fibrinogen, coagulation factors or fibrinolysis to thrombus development. Unlike standard coagulation tests, viscoelastic assays offer a more comprehensive insight on how global coagulation unfolds in vivo.

In the last few decades, VETs have been used to evaluate SIC and sepsis-associated overt DIC, considering the persistent need to predict and prevent poor outcomes and complications in sepsis more efficiently. Up until now, a few conclusions couldbe drawn from research work using VETs in sepsis. Three viscoelastic patterns were found in septic patients—hypocoagulability, normocoagulability and hypercoagulability; the abnormal patterns were identified with a prevalence ranging from 43% to 100% [283]. Hypocoagulable patterns identified in multiple studies, using various protocols, correlated with mortality rates [284,285,286,287]. Certain VET parameters can predict and diagnose DIC in septic patients, particularly in the phase of consumptive coagulopathy [288,289,290,291]. In accordance with the pathogenic sequence of sepsis progression, an early tendency towards hypercoagulation and early “fibrinolysis shutdown” hasbeen identified through VETs in some study groups; these modifications increase morbidity and mortality [240]. Eventually, abnormal VET patterns turned out to be particularly sensible in detecting coagulopathy as they seem to impact prognosis despite near-normal standard coagulation panels [292].

In COVID-19 patients, a hypercoagulable state with accentuated clot kinetics, clot strength and “fibrinolysis shutdown” wasfound through VETs [256,293,294]. The hypofibrinolytic pattern together with high D-dimer levels was associated with an increased risk of thromboembolic events [295]. Unlike sepsis patients, COVID-19 patients rarely display hypocoagulable VET patterns. Further studies are required to determine the prognostic value of VETs for micro- and macrothrombosis in COVID-19 and their possible role in the therapeutic approach of CAC.

## 5. Conclusions

The molecular mechanisms behind the inflammatory and immunological responses of bacterial sepsis, on the one hand, and COVID-19, on the other hand, together with the associated coagulation disorders, reveal multiple similarities between SIC and CAC pathogenesis.

Humoral and cellular actors mediate the common pathways of inflammation–coagulation crosstalk. Both bacterial- and viral-derived PAMPs initiate immune responses following binding to PRRs. Subsequent inflammasome and NF-κB pathway activations are responsible for the initial coagulation activation, mainly via the extrinsic pathway. Cellular effectors are further involved in the unfolding of widespread immunothrombosis. Monocytes represent the main source of circulating TF and subsequent extrinsic pathway enhancement. NETs formation and platelet activation establish excellent structural and functional circumstances for inflammation and thrombosis to potentiateeach other. In addition, ECs dysfunction equally worsens hemostatic and immune disorders in SIC, as well as in CAC.

The resulting clinical picture of SIC and CAC shows, however, some distinctive features. In fact, these two coagulopathies seem to share the same “ingredients” (i.e., molecular mechanisms), but they may not share the same “recipe” (i.e., dynamics). CAC is marked by a pulmonary-localized initial phase of intravascular coagulation due to the important respiratory tropism of SARS-CoV-2. This phase may be followed by a disseminated form of coagulopathy, potentially evolving towards a consumptive pattern.

CAC does not smoothly superimpose over classical SIC and/or DIC definition and criteria. Therefore, proposals of new definitions and/or classifications for CAC should be supported and an effort to clinically validate them encouraged.

## Figures and Tables

**Figure 1 jcm-12-00601-f001:**
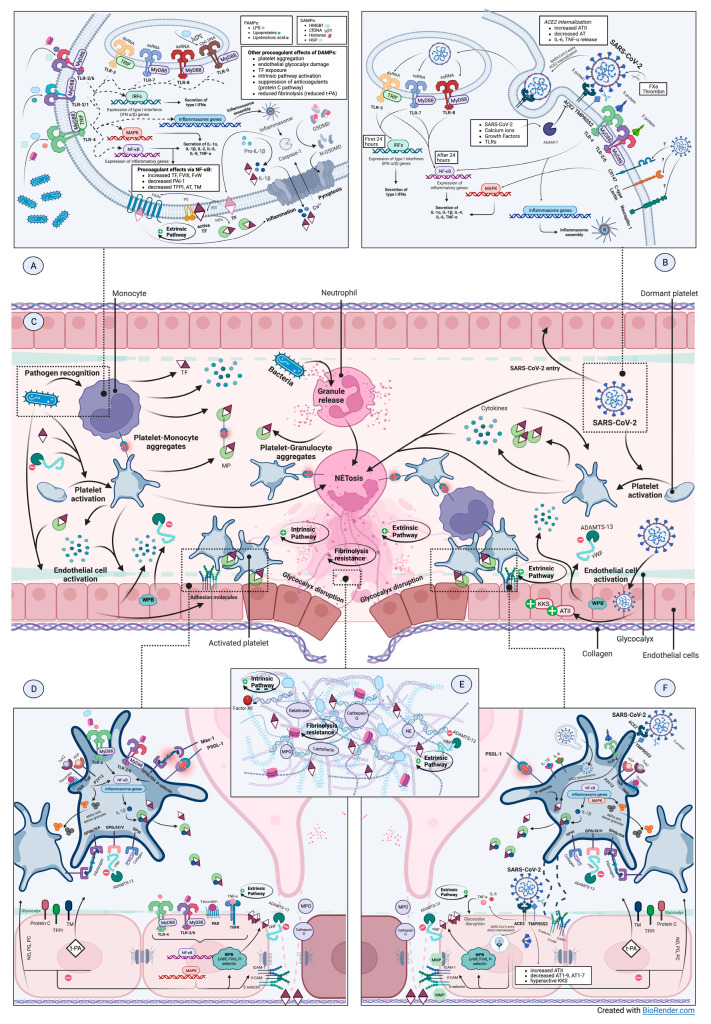
A comparative overview of pathogenic mechanisms involved in sepsis-induced coagulopathy and COVID-19-associated coagulopathy. (**A**). Pathogen recognition pathways in bacterial sepsis. (**B**). Molecular mechanisms of SARS-CoV-2 entry and subsequent cell injury. (**C**). Intravascular interplay of pathogens (i.e., bacteria and SARS-CoV-2), effector cells and humoral factors in the context of sepsis-induced coagulopathy versus COVID-19-associated coagulopathy. (**D**)**.** Platelet and endothelium dysfunctions in bacterial sepsis. (**E**). NETs components and NET-associated prothrombotic status. (**F**). Platelet and endothelium dysfunctions in COVID-19. Abbreviations: ACE2, angiotensin-converting enzyme 2; ADAM17, a disintegrin and metalloprotease 17; ADAMTS13, a disintegrin and metalloprotease with a thrombospondin type 1 motif, member 13; ADP, adenosine diphosphate; AT, antithrombin; ATII/1-9/1-7, angiotensin II/1-9/1-7; CD147, cluster of differentiation 147; Cf/CpG DNA, cell-free/cytosine-guanine pair desoxyribonucleic acid; CLR, C-type lectin receptor; DAMP, damage-associated molecular pattern; FVIII/X/XII, coagulation factor VIII/X/XII; GP-Iba-IX-V, glycoprotein Iba-IX-V; GPIIb/IIIa, glycoprotein IIb/IIIa; GPVI, glycoprotein VI; GSDMD, Gasdermin D; HMGB1, high mobility group box 1; Hsp, heat-shock proteins; ICAM-1, intercellular adhesion molecule 1, IFN, interferon; IL, interleukin; IRF, interferon-regulatory factor; KKS, kallikrein-kinin system; LPS, lipopolysaccharides; Mac-1, macrophage 1 antigen; MAPK, mitogen-activated protein kinase; MMP, matrix metalloprotease; MP, microparticle; MPO, myeloperoxidase; MyD88, myeloid differentiation primary response 88; NE, neutrophil elastase; NET, neutrophil extracellular trap; NF-κB, nuclear factor kappa B; NO, nitric oxide; PAI-1, plasminogen activator inhibitor-1; PAMP, pathogen-associated molecular pattern; PAR, protease-activated receptor; PC, prostacyclin; PDI, protein disulfide isomerase; PG, prostaglandin; PS, phosphatidyl-serine; PSGL-1, P-selectin glycoprotein ligand-1; ss/dsRNA, single-stranded/double-stranded ribonucleic acid; TF, tissue factor; TFPI, tissue factor pathway inhibitor; TLR, Toll-like receptor; TM, thrombomodulin; TMPRSS2, transmembrane protease serine 2; TNF-*α*, tumor necrosis factor *α* TNFR, TNF receptor; t-PA, tissue-type plasminogen activator; TRIF, toll-interleukin 1 receptor-domain-containing adapter-inducing interferon-beta; TxA2, thromboxane A2; TxR, thromboxane receptor; VCAM-1, vascular cell adhesion molecule 1; vWF, von Willebrand Factor; WPB, Weibel Palade Bodies.

**Table 1 jcm-12-00601-t001:** Mechanisms of sepsis and COVID-19-associated coagulopathies.

Commonalities between Sepsis and COVID-19-Associated Coagulopathies	Characteristics of Sepsis-Induced Coagulopathy	Characteristics of COVID-19-Associated Coagulopathy
TLRs and CLRs found on cell membranes and endosomes interact with PAMPs and DAMPs [10,11]. The synthesis of proinflammatory cytokines is activated via MAPK and NF-κB pathways [10,11,209]. Increased PAI-1 levels can be found in both COVID-19 and sepsis. [45,46] TF, FVIII, u-PA, PAI-1, TFPI, antithrombin and thrombomodulin are endpoints of the NF-κB pathway [12,13,14,15,16]. Histones promote platelet aggregation, bind prothrombotic molecules, and damage the antithrombotic properties of the endothelial glycocalyx [50,51]. Activated neutrophils generate prothrombotic and hyperinflammatory states via NETs formation and inflammasome activation, leading to immunothrombosis [88,89,90,91,92,93,94,95,96]. Intercellular interactions between platelets, endothelial cells, neutrophils and monocytes lead to MPs release and TF exposure [145,179]. Monocytes, as a source of soluble TF, promote the activation of coagulation via the extrinsic pathway [72,73]. Active platelets and MPs induce adhesion molecules’ expression and the dysfunction of endothelial cells [127,128,129,130]. The antithrombin pathway and protein C anticoagulant system become dysfunctional due to impaired protein C synthesis and activation via thrombomodulin and endothelial protein C receptor deficiency [52,53,54,226,227]. Tie2, NF-κB and MAPK signaling lead to ECs acquisitionof a proinflammatory and prothrombotic phenotype [207,208,209,210,211,212,213]. IL-6 contributes to vascular permeability and TNF-α worsens the glycocalyx disruption of endothelial cells [219].	HMGB1 can modulate fibrinolysis by interacting with plasminogen and t-PA, can promote coagulation via TF exposure on macrophages and inhibits the protein C pathway [25,26,27,28,29,30,31,32,33,34,35,36,37,38,39]. cfDNA activates coagulation via FXI and FXII [40,41]. High cfDNA concentrations inactivate t-PA by PAI-1 [42,43,44]. Increased TF release from monocytes occurs upon stimulation by LPS [74,75,76]. Fibrinolysis resistance is enhanced via plasminogen altering within NETs, the formation of fibrin–DNA tight complexes insensitive to plasmin and the activation of PAI-1 by cfDNA [42,114,123,124]. Sepsis is associated withADAMTS13 deficiency, which generates large circulating vWF multimers, excessively activating platelets [234,235,236,237]. Platelet activation can arise via direct and indirect bacterial–platelet interactions [153]. A septic proinflammatory status induces (via IL-6 and IL-3 stimulation) the release of thrombopoietin-independent thrombocytes with more numerous TLRs and interleukin receptors, more IL-6 and TNF-α [72,196]. Endothelial cells’ activation can be triggered by TNF-α and thrombin, bacterial LPS and other PAMPs, HMGB1 and other DAMPs, cytokines, such as IFN-γ and IL-1β and shear stress [72,207,208,209,210,211,212,213,214]. Reduced TFPI and t-PA synthesis and increased PAI-1 expression by ECs potentiate the prothrombotic status [179,209,219,224,228]. Glycocalyx injury is initiated by specific enzymes (glucuronidases, hyaluronidases, plasmin, ROS) and hypervolemia (via excessive fluid administration within sepsis management) [227,228,229,230,231].	Activated coagulation FX and thrombin may cleave S protein and promote viral entry in a potential inflammation–coagulation positive feedback loop [61]. Platelet–monocyte interactions are themain stimulus for monocyte activation with robust cytokine and chemokine secretion and TF expression [77]. Low-density granulocytes are found in greater proportions in COVID-19 patients; they are more active in generating NETs, further enhancing hypercoagulability-mediated organ damage [94,95,96]. Platelets from COVID-19 patients exhibit a particular hyperreactivity to low-dose common agonists (such as collagen, α-thrombin or ADP) [143,144,145]. IL-6 and TNF-α can directly activate platelets; IL-6 and IL-1β can prime platelets before stimulation by classical agonists [161]. Pulmonary-residing megakaryocytes are theoretically susceptible to SARS-CoV-2 infection and may transfer viral particles and cytokines when new circulating platelets are generated [144]. SARS-CoV-2 can infect endothelial cells via ACE2 and TMPRSS2, with endotheliitis as a result of direct viral entry [208]. S-protein-related damage increases adhesion molecules’ exposure, ROS synthesis and matrix metalloproteinases’ release with the disruption of the endothelial barrier [179,209,215,216,217,218,219]. COVID-19 induces a hyperactive state of the KKS pathway, promoting vascular permeability [217,220,221]. Excessive complement activation via the lectin pathway accentuates endothelial-derived immunothrombosis [217,220,221].

ACE2—angiotensin-converting enzyme 2; ADAMTS13—ADisintegrinandMetalloprotease with ThromboSpondin type 1 motif, member13; ADP—*adenosine*diphosphate; cfDNA—cell-free desoxyribonucleic acid; CLR—C-type lectin receptor; COVID-19—coronavirus disease 2019; DAMP—damage-associated molecular pattern; EC—endothelial cell; FVIII, FX, FXI, FXII—coagulation factor VIII, X, XI, XII; HMGB1—high mobility group box 1; IFN-γ—interferonγ; IL-1β, IL-3, IL-6—interleukin 1β, 3, 6; KKS—kallikrein-kinin system; LPS—lipopolysaccharides; MAPK—mitogen-activated protein kinase; MP—microparticle; NET—neutrophil extracellular trap; NF-κB—nuclear factor kappa B; PAI-1—plasminogen activator inhibitor-1; PAMP—pathogen-associated molecular pattern; ROS—reactive oxygen species; TF—tissue factor; TFPI—tissue factor pathway inhibitor; Tie2—tyrosine kinase with immunoglobulin-like loops and epidermal growth factor homology domains-2; TLR—Toll-like receptor; TMPRSS2—transmembrane protease serine 2; TNF-α—tumor necrosis factor α; tPA—tissue type plasminogen activator; TxA2—thromboxane A2; u-PA—urokinase-type plasminogen activator; vWF—von Willebrand factor.

## Data Availability

The data presented in this study are available on request from the corresponding authors.

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
