# Peer review of "Coagulation Disorders in Sepsis and COVID-19—Two Sides of the Same Coin? A Review of Inflammation–Coagulation Crosstalk in Bacterial Sepsis and COVID-19"

_jcm, 2023, doi:10.3390/jcm12020601_

Round 1

Reviewer 1 Report

The authors have done an admirable job of summarizing the literature relating to coagulation disorders occurring in Sepsis and COVID-19. However, this narrative review can be improved:

-Several studies have demonstrated that platelet activation due to COVID-19 differs from that occurring in bacterial sepsis. Please cite these references and discuss them.

PMID: 33196292

PMID: 34027292

PMID: 33560374

-Page 10, paragraph: How does SARS-CoV-2 infect cells?. I think it is more relevant to discuss the interaction of SARS-CoV-2 especially with blood cells, red blood cells, white blood cells and platelets, because these cells directly affect hemostasis and coagulation. For example, several laboratories have demonstrated ACE2 expression on the platelet surface while other studies have not found it. Please cite and discuss these results.

-Adding a table summarizing the major commonalities between sepsis and COVID-19 would be an asset.

Author Response

Thank you for giving us the opportunity to submit a revised draft of the manuscript Coagulation disorders in sepsis and COVID-19 – two sides of the same coin? A review of inflammation-coagulation crosstalk in bacterial sepsis and COVID-19 for publication in “Journal of Clinical Medicine”. We appreciate the time and effort you dedicated to provide feedback on our manuscript; we have incorporated all your suggestions in the revised form of the manuscript.

Reviewer 2 Report

Numbers of references are duplicated. Text and figure legend is very difficult to read as there are many numbers that cut out the words and is not clear what those numbers mean. 

The main problem with this article is text about D-Dimers not being associated with severity of disease contradicts many articles that state the opposite. More information about this topic is necessary. 

Author Response

(The authors gave the same response as above.)

Reviewer 3 Report

This is a comprehensive review of SIC and CAC. However, the front half (especially sections 3.1. and 3.2.) are rather summaries of already reported findings. I suggest the authors focus more on the points that they want to emphasize (chapters 4 and 5). As for chapter 3, I suggest not enumerating the findings reported elsewhere but describing the discussion lead from those findings. Then, you can make this review more concise and valuable (also can reduce a significant number of references).

I should avoid redundancy, but when I first looked at the title, I expected this review compares the pathogenesis, clinical features, and management of the coagulopathies arising from sepsis and COVID-19. However, the authors explained the pathogenesis of each disease independently in section 3.1. and 3.2. Many articles have already described the detail of the pathogenesis of SIC and CAC and many readers will be tired of them. Although many of the mechanisms are overlapped, the major players are different, the relevance of innate and adaptive immunity will also be different, and systemic or localized reactions and dynamic changes depending on the diseases’ progress may not be the same. Hopefully, those points are discussed contrastively. The readers want to know which specific points make the difference between SIC and CAC. Since most of the parts of Chapters 4 and 5 are well-written and agreeable, I strongly recommend that the authors reduce the volume of chapter 3. 

Other minor points

The figures are too small and cannot be visible.

Line 74, ‘markers of cellular injury’ should be ‘substances released by cellular injury.

Line 109, there should be a space between TF and has.

Line 147, some reported that PAI-1 increases in COVID-19.

Line 156, not only protein C-thrombomodulin pathway but deterioration of antithrombin pathway is also important.

Line 191, others reported that the MPs from platelets and endothelial cells are predominant. The number of monocytes is smaller than the other cells.

Author Response

(The authors gave the same response as above.)

Round 2

Reviewer 3 Report

The authors responded well.